# Imidazoles as Serotonin Receptor Modulators for Treatment of Depression: Structural Insights and Structure–Activity Relationship Studies

**DOI:** 10.3390/pharmaceutics15092208

**Published:** 2023-08-26

**Authors:** Kapil Kumar Goel, Somesh Thapliyal, Rajeev Kharb, Gaurav Joshi, Arvind Negi, Bhupinder Kumar

**Affiliations:** 1Department of Pharmaceutical Sciences, Gurukul Kangri (Deemed to Be University), Haridwar 249404, Uttarakhand, India; 2Department of Pharmaceutical Sciences, HNB Garhwal University, Chauras Campus, Srinagar Garhwal 246174, Uttarakhand, Indiagarvpharma29@gmail.com (G.J.); 3Amity Institute of Pharmacy, Amity University, Noida 201313, Uttar Pradesh, India; 4Department of Bioproduct and Biosystems, Aalto University, 02150 Espoo, Finland; 5Department of Chemistry, Graphic Era (Deemed to Be University), Dehradun 248002, Uttarakhand, India

**Keywords:** imidazole, drug discovery, antidepressants, serotonin reuptake inhibitors, 5-HT_6_, structure–activity relationship, patent analysis

## Abstract

Serotoninergic signaling is identified as a crucial player in psychiatric disorders (notably depression), presenting it as a significant therapeutic target for treating such conditions. Inhibitors of serotoninergic signaling (especially selective serotonin reuptake inhibitors (SSRI) or serotonin and norepinephrine reuptake inhibitors (SNRI)) are prominently selected as first-line therapy for the treatment of depression, which benefits via increasing low serotonin levels and norepinephrine by blocking serotonin/norepinephrine reuptake and thereby increasing activity. While developing newer heterocyclic scaffolds to target/modulate the serotonergic systems, imidazole-bearing pharmacophores have emerged. The imidazole-derived pharmacophore already demonstrated unique structural characteristics and an electron-rich environment, ultimately resulting in a diverse range of bioactivities. Therefore, the current manuscript discloses such a specific modification and structural activity relationship (SAR) of attempted derivatization in terms of the serotonergic efficacy of the resultant inhibitor. We also featured a landscape of imidazole-based development, focusing on SAR studies against the serotoninergic system to target depression. This study covers the recent advancements in synthetic methodologies for imidazole derivatives and the development of new molecules having antidepressant activity via modulating serotonergic systems, along with their SAR studies. The focus of the study is to provide structural insights into imidazole-based derivatives as serotonergic system modulators for the treatment of depression.

## 1. Introduction

The discovery of serotonin dates back to the mid-19th century, when scientists explored the physiological implications of numerous chemical messengers in the body [1]. Serotonin was first isolated from the blood serum in 1937 by Vialli and Erspamer, Italian scientists. The group identified the highest reservoir of serotonin in the enterochromaffin cells of the gut system. Two other scientists, Rapport and Page, further crystallized the chemical and coined the name ‘serotonin’ in 1945. They and the former scientists found that this very chemical was capable of causing smooth muscle contraction. Owing to this, they named this chemical “serotonin,” derived from two words, ‘sero’ meaning presence in blood serum, and ‘tonin,’ meaning the ability to induce contraction in smooth muscles [2,3].

Additional research revealed that serotonin high-concentration reservoirs are also located in platelets and the central nervous system, in addition to the enterochromaffin cells of the gut [4]. This chemical was also discovered to modulate numerous physiological processes other than smooth muscle contraction and was found to be involved in regulating mood, sleep, appetite, and pain perception [5]. Later, as the research progressed, scientists were keen to discover the specific receptors through which serotonin exerts its physiological effects. It was in the 1970s that scientists discovered two significant classes of serotonin receptors, viz., 5-HT1 and 5-HT2, that were further divided into numerous subtypes [6]. The discovery of these receptors and their subtypes led to advancements in understanding their molecular biology, unraveling numerous physiological questions associated with their signaling [7]. The research also facilitated the scientist’s ability to explore the impact of serotonin in various pathological and physiological conditions and accordingly develop specific serotonin modulators. These modulators include but are not limited to selective agonists and antagonists, selective serotonin reuptake inhibitors (SSRIs), for treating associated conditions that include mental health conditions, including depression and anxiety disorders [8].

With the advancement in molecular neuroscience in the last 60 years, there has been a paradigm shift in developing antidepressant-based therapies [9]. Numerous studies are underway that attempt to recognize the other orchestrating partners with serotonin in combination with or independently associated with depression. Besides paving the way for new research in depression, the studies also expressed their concerns over the serious fears among the patients prescribed serotonin modulators and overmedicalized for years [10]. The US FDA approves numerous modulators for treating any imbalance in serotonin physiology, including (a) Selective serotonin reuptake inhibitors (SSRIs): Citalopram, Escitalopram, Fluoxetine, Fluvoxamine, Paroxetine, Sertraline, Vilazodone, and other drugs; (b) Serotonin and norepinephrine reuptake inhibitors (SNRIs): Venlafaxine, Duloxetine, Desvenlafaxine, and other drugs; (c) Tricyclic and tetracyclic antidepressants: Amitriptyline, Imipramine, Nortriptyline, Doxepin. Notably, more than 90% of these drugs belong to the nitrogen heterocyclic categories, which mainly include indole rings (Fluoxetine, Sertraline, Vilazodone, Vortioxetine); Benzimidazole (Vortioxetine); Benzazepine (Tianeptine); Benzothiazepine (Esmirtazapine); and other drugs [11].

Among the known nitrogen-containing heterocycles, imidazole is an essential heterocycle moiety explored for its biological and medicinal attributes. Imidazole is a widely explored five-membered aromatic heterocyclic compound found in synthetic and natural compounds [12]. Imidazole-containing molecules attach to a wide range of therapeutic targets thanks to their unique structural characteristics and electron-rich environment, resulting in a wide range of bioactivities [13,14,15].

The imidazole ring, as a heterocycle, is part of essential amino acids, including histidine (a histamine precursor). It is widely used in drugs such as antifungal agents (ketoconazole, clotrimazole) [16], antihistamines (cimetidine) [15], COX-inhibitors [17], as well as in other diseases (for example, anticancer [18], antibacterial, antitubercular [19], anti-inflammatory [20], antineuropathic [21], anti-Alzheimer [22,23], antihypertensive [24], antiviral [25], anti-obesity [26], and antiparasitic activity [27]). Significantly, due to facile structural substituting availability on the imidazole ring, various derived molecules were exploited for gaining potent anticancer activities, for example, topoisomerase inhibitors (as imine-amide imidazole conjugates targeting liver and lung cancer cells) [28]. Additionally, they have been employed in naturally derived compounds, such as MIM1 [29], Meiogynins [30,31,32], and synthetically designed Mcl-1 inhibitors (as in the form of imidazolidine-2,4-dione for chronic myelogenous leukemia cells, prostatic cancer cells, and breast cancer cells [33,34]). Furthermore, these compounds have been investigated for their potential in targeting insulin-linked cancer cells (insulin receptor-A and IGF-1R and their heterodimers [35,36]). They have also been explored for hormonal-based targeting of cancer cells (estrogen-based drugs targeting breast cancer [37,38,39,40,41]).

The application of imidazole rings is not limited to small molecule-based medicinal chemistry; aspects of coordination chemistry and targeted synthesis (with chemical biology applications) were explored as well. For example, medicinal coordination chemistry utilizing elements of supramolecular chemistry has recently gained interest [42]. In coordination chemistry, the imidazole ring presented broad applications with its superlative chemical feasibility, such as lone pairs of nitrogen in imidazole being available to facilitate making coordinate bonds with central metals, as observed in nature (in some of the essential biomolecules, hemoglobin, and hemocyanin), encouraging researchers to develop metal-containing medicinal compounds. The imidazole derivatized coordinate compounds additionally exhibit more advantages, such as lower incidences of cellular resistance, increased therapeutic efficacy, selective cellular targeting [13], and probing, as broadly summarized: (A) Imidazole-based supermolecules as anticancer agents (alkylating agents [43], noble metal complexes with anticancer activities [44,45], light-activated cytotoxicity of ruthenium-metal complexes [46]). (B) Imidazole-based coordinate compounds for organelles and cellular detection: (a) Boron pyridyl imidazole complex with a characteristic of twisted intra-molecular charge transfer (TICT) does specific detection of BSA levels (identifying denatured BSA versus native BSA) [47]; (b) Imidazole containing dinuclear Ru (II) complex is a lysosome-specific probe, accumulating (specifically) in the lysosomes and therefore assisting in identifying the HeLa cells from healthy HEK293 cells [48]; (c) Gram-negative bacteria differentiated from Gram-positive bacteria using imidazole containing dibenzimidazole-substituted pyridine [49]. (C) Cellular biomolecular and metal detection and probing: (a) Imidazole containing diphenyl derivatives [50] and anthraquinones detected the biological mercaptans (cysteine, homocysteine, and glutathione) levels [51]. (b) Imidazole containing fluorescent sensors detected approximately 4.70 × 10^−7^ mol/L adenine levels [52]. (c) Fluorescent probe-conjugated imidazole–pyridine pharmacophore structures demonstrated high sensitivity at a molar concentration of 3.38 μM for Ag (I), applicable to liver cellular imaging [53]. (d) Thiophene-derivatized imidazoles as on–off fluorescent reversible chemosensors detecting intracellular Pd (II) ions in living cells (to a level of 20 μg/mL) [54]. (e) Fluorescent probe-based peptide receptors detected intracellular Cu (I) ions in the Golgi apparatus (even in living A549 cells) [55]. (f) Tripyridyl imidazole molecule as a dual sensing probe of Hg (II) (pH 6–8) and Cu (II) (pH 3–11) ions detecting at 0.77 and 1.58 μM, respectively [56]. (g) Ruthenium (II) complexes containing imidazole as ^1^O_2_-responsive fluorescent probes as an “on–off”-type fluorescent pH sensor (pK_a1_ = 1.12 ± 0.15, pK_a2_ = 6.90 ± 0.24, pK_a1_* = 1.09, and pK_a2_* = 6.92) [57]. (h) Fluorescein dye containing imidazole as a colorimetric and fluorescent chemical sensor for fluoride detection with an application in live cells [58]. (i) Cu (II) complexed Bipyridyl imidazole derivative selectively detected the HS^−^ ions and was used to develop a fluorescence microplate assay [59]. (j) Fluorescent zinc complexes presented a high-potential HS^−^/H_2_S fluorescence sensor and detection probes at cellular levels [60]. (k) Ir (III) containing a methylene-bridged benzimidazole-substituted complex displayed a high selectivity for pyrophosphate ions (H_2_P_2_O_7_^2−^) with lower cellular HeLA cell toxicity [61]. (l) Fluorescent-active benzimidazole derivatives have high selectivity toward aqueous solubilized Ag (I) within (<30 s) [62]. (m) A reversible naphthalimide-based probe detected Hg (II) ions in a phosphate buffer over a wide pH range (7.0–10.0), which is highly applicable in biological experiments [63]. (n) Imidazolyl Schiff bases exhibit Zn (II) detection as low as 6.78 × 10^−9^ M compared to the recommended sensitivity detection according to the World Health Organization’s drinking water guidelines (7.6 × 10^−5^ M) [64]. (o) Imidazole-based anthracene structure detection limit of Zn (II): 1.0 × 10^−9^ M [65]. (p) Tridentate dibenzimidazole–pyridines have a detection limit of 3.09 × 10^−7^ M toward Zn (II) ions [66]. (q) Imidazole-based supermolecules as probes for Cu (II) detection (allyl-substituted imidazole derivative with a detection limit for cupric ion (Cu^2+^) of 1.01 nM [67]; dibenzimidazole derivative with a detection limit of 0.094 μM within a 1 s time period [68]; tetraphenyl ethylene-functionalized aryl imidazole-derivative detection limit of aqueous solubilized Cu (II) at 34.8 nM [69]). (r) Imidazole-based iron chemosensors (fluorescence “on–off” aniline-derived imidazole probe demonstrated high selectivity and sensitivity for ferric ions Fe (III) with a detection limit of 0.72 μM/L at 30 min) [70].

Several imidazole-based medicines have been widely employed in clinical trials to treat various disorders with significant therapeutic promise. Imidazole-containing drug research and development is becoming a more active and appealing issue in medicinal chemistry due to its substantial therapeutic usefulness. Therefore, this article is kept forth to illuminate the landscape of imidazole-based drug discovery and development, focusing on structure–activity relationship aspects (SAR) against serotonin receptors to target depression. The insights covered will compel medicinal chemists, synthetic chemists, biologists, and pharmacologists to explore and unravel this exciting area for the drug discovery of serotonin modulators.

### Etiology of Depression, Structural and Mechanistic Insights

According to the World Health Organization, depression is a severe condition affecting millions of people worldwide and is one of the primary causes of disability [71,72]. Monoamine neurotransmitters like norepinephrine and serotonin (5-hydroxytryptamine, 5-HT) have been used as significant indicators of psychiatric disorders, such as anxiety and depression, for several decades [73,74]. Serotonin exists in both the CNS and PNS systems, which describe its nature as both an autacoid and a neurotransmitter [75,76]. It is released by serotonergic neurons. The serotonergic system is linked to the regulation of mood, emotion, and sleep, as well as a variety of behavioral and physiological activities [77]. It is one of the most studied and multifunctional biogenic amines among neurotransmitters. Depending on the physiological action, occurrence, agonist, and antagonist, it is categorized into various classes, among whom 5-HT_1_ (5-HT_1A_, 5-HT_1B_, 5-HT_1D_, 5-HT_1E_, and 5-HT_1F_), 5-HT_2_ (5-HT_2A_, 5-HT_2B_, and 5-HT_2C_), 5-HT_3_, 5-HT_4_, 5-HT_5_ (5-HT_5A_, 5-HT_5B_), 5-HT_6_, and 5-HT_7_ have been widely reported in the literature [78]. Serotonin is primarily found in three cell types: (a) serotonergic neurons in the central nervous system and the intestinal myenteric plexus [79], (b) enterochromaffin cells in the gastrointestinal tract mucosa, and (c) blood platelets [80].

Serotonin receptors are categorized as G protein-coupled receptors (GPCRs), which orchestrate the signaling mechanism of serotonin. Serotonin binds to its chief receptors (5-HT1 and 5-HT2 subtypes) within the transmembrane region of the receptor. This region comprises transmembrane helices classified as TM1–TM7 [81]. These transmembranes are further connected via intracellular and extracellular loops, forming a binding pocket that facilitates the formation of serotonin and its competitive modulators. These transmembrane domains are further surrounded by cholesterol molecules that, besides contributing to the receptor’s shape, are also contributing factors for receptor stability and proper receptor folding. The cholesterol composition within the cell membrane directly modulates receptor activity.

The essential amino acids that play a critical role in serotonin and ligand are embedded within the binding pocket. Considering the 5-HT1 subtype receptor, the key amino acid is aspartate (ASP), located in TM3 (ASP116). This amino acid mediates serotonin affinity via the formation of a salt bridge with the amine functionality of serotonin, thus stabilizing the ligand-receptor complex. Considering the 5-HT2 subtype, the key amino acid that is involved in stabilizing the ligand affinity is asparagine (ASN), located in TM7. This specific amino acid is known to interact with the carboxyl group of serotonins via the formation of the H-bond. Apart from these vital amino acids, Arginine (ARG) residues present in TM3 (ARG 134) and TM6 are also found to interact with carboxyl groups via H-bonding and electrostatic interactions. Glutamate (GLU), present in TM2 and TM7, is involved as a proton acceptor or donor required during protein interactions, besides participating in H-bonding and electrostatic interactions with ligands. Another amino acid, histidine (HIS), present in TM3 and TM5, is also associated with receptor activation and participates in ligand stability via the formation of an H-bond, a salt bridge, or by acting as a proton donor or acceptor. In addition to this, cysteine (CYS) residues present in TM2 and TM7 are involved in the stability of the receptor protein via the formation of disulfide bridges. They are vital for ligand recognition and receptor activation. The tryptophan (TRP) residues (particularly TRP 358) available at TM6 and TM7 also contribute to receptor activation along with receptor conformational stability and ligand binding primarily via hydrophobic interactions. Apart from amino acids, three water molecules (W1–W3) within the active domains are also vital in maintaining the stability and activation of the receptor. These water molecules interact with 5-HT (apoprotein), where W1 interacts with the hydroxy group and W2 with the indole ring, followed by the interaction of the primary amine with W3. W3 also interacts with ASP116 by forming an H-bond that is conserved in GPCRs (aminergic), whereas W2 is found to toggle the amino acid residue TRP358, which determines receptor activation [82].

Mechanistically, serotonin and norepinephrine are released in the synaptic cleft, where they activate the postsynaptic cleft and some reuptakes through the pump, where MAO breaks them and moves them back to the presynaptic neuron [83]. In the case of depression treatment using SSRIs/SNRIs as per the monoamine hypothesis, the therapeutic benefits are based on increasing low serotonin levels and norepinephrine (Figure 1) [84,85]. As the name indicates, SSRIs/SNRIs function by preventing serotonin/norepinephrine reuptake and thereby increasing activity. SSRIs block the serotonin transporter (SERT) at the presynaptic axon terminal [86], whereas SNRIs block the norepinephrine reuptake transporter [87]. The information-sending presynaptic cell in the brain releases neurotransmitters into the gap [88]. The neurotransmitters are subsequently identified by receptors on the recipient postsynaptic cell’s surface, which then relay the signal in response to the stimulus. More serotonin (5-hydroxytryptamine, or 5HT) remains in the synaptic cleft when SERT is blocked, which can stimulate postsynaptic receptors for extended periods [89]. Further, SNRIs and SSRIs boost serotonin and norepinephrine levels in the brain. Neurotransmitters, such as serotonin and norepinephrine, are chemical messengers that transfer messages from one region of the body to another. Following the transmission of a signal by neurotransmitters, cells in the brain typically take up these substances and store them for later use. SNRIs and SSRIs block serotonin and norepinephrine reuptake, resulting in higher levels of serotonin and norepinephrine in the synaptic cleft [90].

## 2. Modern Synthetic Methods for Substituted Imidazole Derivatives

Imidazole is reported to exhibit a broad range of applications in pharmaceutical and industrial applications [14]. For example, this organic framework is sought in many drug pharmacophores, such as angiotensin II inhibitors, anti-inflammatory agents [91], anticancer agents, and building blocks of naturally occurring products [13,92]. The imidazole ring is a well-observed ligand in metalloenzymes, and its imidazolium salts are also well-exploited to serve as excellent precursors of stable carbene ligands in various metal complexes [93,94]. The application of imidazolium salts to environmentally friendly ionic solvents is another example [95]. Thenrajan and coworkers explored the role of imidazolate-based bimetallic nickel-iron zeolitic fibers as sensors for serotonin neurotransmitters [96]. Various advancements reported in the synthesis of new imidazole derivatives with various bioactivities using different catalytic systems are described in Figure 2.

The use of metallic catalysts for imidazole synthesis has increased in recent years. This can be attributed to the improved percentage yields, lessening the time required for reaction, and ease of removal from a reaction mixture that make the technique appealing [93,97]. Because of their nontoxic, affordable, reusable, and eco-friendly properties, zinc (Zn)-based heterogeneous catalysts were well-exploited in a range of multicomponent reactions for synthesizing this organic framework. Marzouk et al. [98] developed a one-pot multicomponent synthesis of 1,2,4,5-tetrasubstituted imidazoles via reacting aromatic aldehydes, benzil, 1-amine-2-propanol, and ammonium acetate in the presence of nanoparticles of ZnFe_2_O_4_ catalyst. After 30–50 min, the condensation process with a metal catalyst yielded 87–96% multi-substituted imidazoles. For the production of substituted imidazole, Nejatianfar et al. [99] proposed a magnetically separable nanocatalyst based on copper (II) immobilized on guanidine epibromohydrin-functionalized c-Fe_2_O_3_@TiO_2_ (c-Fe_2_O_3_@TiO_2_-EGCu(II)) with a core–shell structure. Eidi et al. [100] used benzil, substituted aldehydes, and ammonium acetate in a condensation procedure to produce 2,4,5-trisubstituted imidazole conjugates. The best reaction conditions were discovered using 10 mg of catalyst and a 60 percent rate power of ultrasonic irradiation at 40 °C in ethanol. The method’s advantages include recovering the catalyst using an external magnetic field and reusing it for up to five runs without losing activity. Maleki et al. [101] developed a greener synthetic strategy for the formation of 2,4,5-trisubstituted imidazoles via condensation of 1,2-diketone, aromatic aldehydes, and ammonium acetate in the presence of mixed oxide (Fe_3_O_4_/SiO_2_) nanocatalyst, yielding up to 95%. Compared to traditional catalysts such as Fe_3_O_4_, the reaction utilizing Fe_3_O_4_/SiO_2_/urea nanoparticles took 50 min and yielded 95% trisubstituted imidazoles. However, designing and synthesizing trisubstituted scaffolds remain of keen interest among researchers [102,103]. These scaffolds provide a highly functional multitargeting scaffold [104,105], including dendrimers [106,107]. Girish et al. [108] used ZrO_2-_supported b-cyclodextrin as a reusable solid catalyst to synthesize 2,4,5-trisubstituted imidazoles and 1,2-disubstituted benzimidazoles under solvent-free conditions. Using a 40 mol% ZrO_2_-b-Cyclodextrin catalyst, the reaction was screened with various solvents, including water, DMF, ethanol, and solventless systems. The reaction went off without a hitch in a solvent-free environment, and the product was produced with good yields. Using ZrO_2_ nanoparticles as a reusable catalyst, Bajpai et al. [109] investigated the one-pot synthesis of multi-substituted imidazoles. The authors used isatin, aromatic aldehydes, and ammonium acetate as reactants in the presence of 15 mol% ZrO_2_ nanoparticles in a solvent-free environment to create new imidazoles.

Fang et al. [110] described a new method for cyclizing amido-nitriles to produce disubstituted imidazoles. The reaction conditions were moderate enough to include aryl halides, aromatic and saturated heterocycles, and other functional groups. The necessary 2,4-disubstituted NH-imidazoles were obtained by nickel-catalyzed addition to nitrile, followed by proto-demetallation, tautomerization, and dehydrative cyclization. Combining a C2–N3 fragment with an N1–C4–C5 unit has recently been examined as a two-bond disconnection for synthesizing imidazoles. For example, Shi et al. [111] employed this disconnection to make trisubstituted NH-imidazoles in the presence of zinc(II) chloride by reacting benzimidates with 2H-azirines. For the synthesis of 2-aminoimidazoles, Man et al. [112] utilized an approach where vinylazides were transformed in situ into 2H-azirines, which then interacted with cyanamide to create the required 2-aminoimidazoles in moderate to good yields under a range of conditions. Nitriles have also been employed as reagents in metal-free processes to synthesize substituted imidazoles with two bonds. Harisha et al. [113], for example, recently reported the formation of tri-substituted NH-imidazoles by reacting -azidoenones with substituted nitriles. In the absence of a metal catalyst, imidamides can be employed as starting materials for synthesizing imidazoles. In the presence of trifluoroacetic acid, Tian et al. [114] reported the synthesis of substituted imidazole by reacting imidamides with sulphoxonium ylides. At the first, second, and fourth positions, the resulting imidazoles were replaced.

## 3. SAR of Various Imidazole Derivatives

As a new generation of selective 5-HT_7_ receptor agonists, Hogendorf et al. developed and synthesized a series of 27 fluorinated 3-(1-alkyl-1H-imidazol-5-yl)-1H-indoles. By optimizing the halogen bond formation with Ser5.42 as the expected partner, a powerful and drug-like agonist, compound **1**, 3-(1-ethyl-1H-imidazol-5-yl)-5-iodo-4-fluoro-1H-indole (*K_i_* 5-HT_7R_ = 4 nM), was discovered. Excellent water solubility, good selectivity over related CNS targets, high metabolic stability, oral bioavailability, and minimal cytotoxicity were all attributes of the molecule. After i.p. (2.5 mg/kg) treatment in mice, rapid absorption into the blood, a medium half-life, and a high peak concentration in the brain (C_max_ = 1069 ng/g) were discovered. The antinociceptive effect reported in a mouse model of neuropathic pain suggests that **1** might be a long-sought tool chemical in the research of 5-HT_7_ receptor function, as well as a potential analgesic [115]. The detailed SAR analysis of this series of compounds is discussed in Figure 3.

As non-sulfonamide 5-HT_6_ receptor ligands, a new series of 3H-imidazo[4,5-*b*]pyridine and 3H-imidazo[4,5-*c*] pyridine derivatives were reported by Vanda et al. Compound **2** (2-ethyl-3-(3-fluorobenzyl)-7-(piperazin-1-yl)-3H-imidazo[4,5-*b*]pyridine) was identified as a strong 5-HT_6_ receptor partial inverse agonist in G_s_ signaling (K_i_ = 6 nM, IC_50_ = 17.6 nM) after in vitro testing. Compound **2** had good metabolic stability, a favorable cytochrome P450 isoenzyme profile (2D6, 3A4), did not impact PgP-protein binding, and had no mutagenesis effects. The bioavailability and blood-brain barrier (BBB) permeability of this compound further favors its candidature as an investigative molecule in clinical studies. Along with non-neurotoxicity, the use of a synergistic combination of inactive doses of compound **2** (0.1 mg/kg) and donepezil (0.3 mg/kg) to reverse scopolamine-induced memory deficits produced a synergistic effect. In the binding pocket of ligands for 5-HT_6_R, various small or bulky alkyl/aryl substituents were employed at the C2-position of the imidazo[4,5-*b*]pyridine moiety, and SAR studies were developed as discussed in Figure 4 [116].

Zagorska et al. developed and performed pharmacological screening of 2-fluoro and 3-trifluoromethylphenylpiperazinylalkyl derivatives of 1H-imidazo[2,1-*f*]purine-2,4(3*H*,8*H*)-dione (**3a**–**u**) for their antidepressant activity targeting serotonin (5-HT_1A_/5-HT_7_) receptors as well as phosphodiesterase (PDE4B and PDE10A). Although the results from in vitro studies revealed the synthesized compounds as potent 5-HT_1A_, 5-HT_7_, or dual 5-HT_1A_/5-HT_7_ receptor ligands, they had lower inhibitory potencies for PDE4B and PDE10A. Most of the compounds displayed selective 5-HT_1A_ receptor affinity, which encouraged further preclinical investigation. Target compounds **3a**–**u** was investigated for their metabolic stability and lipophilicity properties utilizing micellar electrokinetic chromatography (MEKC) technology and a human liver microsomes (HLM) model, which presented the target compounds as having moderate pharmacophoric features. During FST, compound **3i,** 8-(5-(4-(2-fluorophenyl)piperazin-1-yl)pentyl)-1,3,7-trimethyl-1H-imidazo[2,1-*f*]purine-2,4(3*H*,8*H*)-dione, was found to possess the most prominent antidepressant activity at dosages of 2.5 mg/kg and 5 mg/kg. At a dosage of 2.5 mg/kg, it also showed an anxiolytic effect in the four-plate method. Molecular docking studies also revealed that fluorinated arylpiperazinylalkyl derivatives of 1H-imidazo[2,1-*f*]purine-2,4(3*H*,8*H*)-dione have major pharmacophoric features for the development of the antidepressant and anxiolytic compound. The SAR studies of this series of compounds are discussed in Figure 5 [117].

Arylpiperazine-substituted imidazole derivatives are one of the major classes explored in various neurological disorders [118,119,120]. Zagόrska et al. synthesized a novel series of arylpiperazinylalkyl purine-2,4-diones (**4a**–**u**) and purine-2,4,8-triones (**5a**–**h**) and screened them in vitro for their serotonergic and dopaminergic receptor affinity. Compounds containing an imidazole ring with purine-2,4-diones in their structure have a high affinity for serotonin receptors (5-HT_1A_ and 5-HT_7_) as well as dopamine receptors (D_2_) as compared to compounds with purine-2,4,8-trione. The resultant compounds displayed various CNS activities. During in vivo evaluation, it was discovered that compounds **4h** and **4o** operate as potential antidepressants, while compounds **4r** and **4u** act as both potential antidepressants and anxiolytic agents. It has been demonstrated by molecular docking studies that the presence of 1,3-dimethyl-(1*H*,8*H*)-imidazo[2,1-*f*]purine-2,4-dione at position C7 is required for receptor affinity and selectivity, particularly for 5-HT_1A_ and 5-HT_7_. The SAR studies are discussed below in Figure 6 [121].

The same research group reported two new series of N-8-arylpiperazinylpropyl derivatives of 1,3-dimethyl-(1*H*,8*H*)-imidazo[2,1-*f*]purine-2,4-dione (**6a**–**i**) and amide derivatives of 1,3-dimethyl-6,7-dihydroimidazo[2,1-*f*]purine-2,4-(1H,3H)-dione-7-carboxylic acid (**7a**–**c**) and evaluated for their antidepressant potential using the FST model. During in vitro studies, compound **6h** displayed the highest affinity for 5-HT_1A_ receptors and a K_i_ value of 5.6 nM with high selectivity over 5-HT_2A_ receptors. Further, in vivo evaluation of antidepressant activity revealed that compounds **6a**, 8-[3-(N4-phenyl)-piperazin-N1-yl-propyl]-1,3-dimethyl-(1*H*,8*H*)-imidazo[2,1-*f*]purine-2,4-dione and **6b**, 8-[3-(N4-20-metoxyphenyl)-piperazin-N1-yl-propyl]-1,3-dimethyl-(1*H*,8*H*)-imidazo[2,1-*f*]purine-2,4-dione possesses the most significant activity and reduces the immobility time in FST, similar to the drug imipramine. These long-chain arylpiperazine derivatives with a tricyclic moiety can be further explored for the development of new antidepressant compounds with minimal to no side effects. The SAR studies of these imidazole-based compounds are discussed in Figure 7 [122].

Tokgoz et al. reported a series of benzazole derivatives (**8a**–**h**) and evaluated them for their antidepressant-like activities. Novel benzazole subordinate mixtures were synthesized by the reaction of respective 2-(benzazol-2-ylthio)acetohydrazide and 4-substituted benzaldehydes. The antidepressant-like activity of these compounds was assessed by TST and modified constrained swimming tests (MFST). During in vivo studies, at a dosage of 50 mg/kg, compounds **8a**, **8b**, **8e**, and **8f** (Figure 8) significantly reduced the immobility time in both TST and MFST. There was no change observed in the climbing duration, which demonstrates the selective antidepressant-like action of these compounds. Further, when pre-treated with serotonin synthesis inhibitors, i.e., p-chloro-phenylalanine methyl ester, NAN-190 (a 5-HT_1A_ antagonist), ketanserin (a 5-HT_2A/2C_ antagonist), and ondansetron (a 5-HT_3_ antagonist), these compounds showed opposite effects in mice, which shows that these compounds act via the serotonergic system for antidepressant-like activity. In addition, the locomotor exercises of the creatures were surveyed by an action confinement contraption [123].

Czopek et al. designed imidazolidine-2,4-dione derivatives using a computational ligand design approach for finding new dual-targeting (5HT_1A_ receptor and serotonin transporter (SERT)) agents. Structural optimization of the initial 5HT_1A_R ligands gives a series of Mannich bases (**9a**–**h**) with an aryl substituent at the 5-position of the imidazolidine-2,4-dione as lead molecules, having the best match with the SERT binding site too. However, the in vitro results were found to be contrary to the expected results for this newly synthesized series. Only compounds with the substituent 3-chlorophenylpiperazine (**9c** and **9g**) showed significant affinities for both 5HT_1A_R and SERT, with K_i_ values of 80 nM and 166 nM (**9c**) and 76 nM and 278 nM (**9g**), respectively. The results demonstrated the high importance of imidazole substituted with an arylpiperazine moiety. The selected compounds **9c** and **9g** further showed partial agonist and antagonist properties at the 5HT_1A_ and 5HT_2A_ receptor sites, respectively, and had low affinity for α_1_ receptors. Such encouraging results further motivated the evaluation of the compounds for their antidepressant and anxiolytic properties. The studies demonstrated that compounds **9c** and **9g** significantly reduced the immobility time in mice in FST, and the antidepressant effect of **9c** was found to be comparable to the reference drug imipramine. The compounds at the doses administered did not affect locomotor activity. The detailed SAR study of these compounds is discussed in Figure 9 [124].

Seo et al. investigated a series of imidazole cores containing arylpiperazine-4-carboxamide derivatives targeting 5-HT_2A_ receptors and 5-HT transporters for the treatment of depressive disorders. Aa imidazole moiety provides good pharmacokinetic properties in drug development. The research group focused on developing core molecules around this moiety. The current series of compounds displayed potential IC_50_ values against 5-HT_2A/2C_ and serotonin reuptake inhibition. Further, these compounds showed significant in vivo antidepressant-like effects in the FST. Based on the evaluation, compounds **10**, **11**, **12**, **13**, and **14** (Figure 10) were found to be more promising for antidepressant activity, and it was concluded that this imidazole series could be used as a promising tool for the development of new antidepressants [125].

One of the imidazole-based novel neuronal nitric oxide synthase (nNOS) inhibitors, i.e., 1-(2-trifluoromethylphenyl)-imidazole (**15**, Figure 10), is sought to synergize the action of other antidepressants via the serotonergic system. Ulak and colleagues reported that FST models augment the antidepressant activity of TCAs like imipramine, SSRIs like citalopram and fluoxetine, and the selective serotonin reuptake enhancer tianeptine. Unfortunately, the same results were not observed with the noradrenaline reuptake inhibitor reboxetine, which confirms the action of **15** via the serotonergic system [126]. Sherwin et al. further confirmed the effect and regional-specific modulation of 1-(2-trifluoromethylphenyl)-imidazole (**15**) [127].

## 4. Other Imidazole-Based Serotonin-Modulating Agents

In continuation of efforts to develop new serotonin reuptake inhibitors, Lauro and coworkers recently (2022) reported strychnidin-oxiran-napthalenol derivatives **16** and **17** (Figure 11), which were found to possess dual noradrenaline and serotonin reuptake inhibitory activity in docking models [128]. Although these results were only based on computational docking studies, there is no further proof regarding their clinical efficiency.

Czopek et al. synthesized a new series of compounds derived from 4-methoxy-1H-isoindole-1,3(2H)-dione as dual targeting ligands having affinity for serotonin receptors along phosphodiesterase 10A (**18a**–**t**). To understand the structure–activity relationship of compounds, 4-methoxy-1H-isoindole-1,3(2H)-dione derivatives (**18a**–**t**) conjugated with a variety of amine moieties, including imidazole and benzimidazole, were screened against both targets. In this study, compounds having a benzimidazole moiety conjugated (**18g**) (2-[4-(1H-benzimidazol-2-yl)butyl]-4-methoxy-1H-isoindole-1,3(2H)-dione) were found to possess the most balanced profile with affinity toward both targets based on in vitro experiments. The lead compound **18g** was investigated for its safety profile and subjected to computational studies, where it should have good affinity and bind at the active site, with such interactions responsible for the inhibition of the PDE10A enzyme. The SAR studies of the current series of compounds against serotonin receptors are described in Figure 12 [129].

Zmudzki et al. investigated the influence of the modifications of the fused imidazole nucleus (xanthine) and the effect of the substituent in position eight on the affinity for serotonin 5-HT_1A_, 5-HT_2A_, 5-H_T6_, 5-H_T7_, and dopamine D_2_ receptors. For this purpose, this group reported different series of arylpiperazynylalkyl derivatives of 8-amino-3,7-dimethyl-1H-purine-2,6(3H,7H)-dione (total 26 compounds). They performed screening of these compounds in vitro against 5-HT_1A_, 5-HT_2A_, 5-H_T6_, 5-H_T7_, and dopamine D_2_ receptors. This study aimed to create a thorough structure–activity relationship profile via various substitutions at position eight with eight different amino moieties. In preliminary evaluation parameters, compounds **19**–**23** (Figure 13) were selected as promising leads, which were further subjected to functional assays for the 5-HT_1A_ and D_2_ receptors. The results demonstrated that these arylpiperazynylalkyl derivatives of 8-amino-3,7-dimethyl-1H-purine-2,6(3H,7H)-dione act as potential antagonists of 5-HT_1A_ receptors while having agonistic, partial agonistic, or antagonistic activity for D_2_ receptors. The SAR analysis revealed that the lipophilic substituent at the 8th position plays a crucial role in affinity toward various serotonin receptors (especially 5-HT_1A_, 5-HT_6_, and 5-HT_7_). The removal of the lipophilic moiety caused a complete loss of affinity toward these receptors. Further, the compounds with lipophilic substituents, such as propoxy or N-ethylbenzylamino substituents, possess the most optimal affinity, which was further joined by dipropyloamino and piperidine-1-yl substituents due to their occupancy of similar volumes and similar lipophilic behavior [130].

Hogendorf et al. utilized aromatic basic groups like imidazoles or thiazoles as aminergic receptor ligands and designed compounds from the N-(1*H* imidazol-2-yl)acylamide chemotype (**24a**–**t** and **25a**–**d)**. The synthesized compounds displayed high affinity for 5-HT_6_R and high selectivity over 5-HT_1A_, 5-HT_2A_, 5-H_T7,_ and D_2_ receptors. In this series, compound 4-methyl-5-[1-(naphthalene-1-sulfonyl)-1H-indol-3-yl]-1*H*-imidazol-2-amine (**24i**) showed the reversal of scopolamine-induced cognitive impairment in rats. The replacement of the amine group of the 5-HT_6_ receptor with 2-aminoimidazole, which is its bioisostere, resulted in highly potent compounds with variable physicochemical behavior. Several studies were performed to analyze the basic character of compounds, demonstrating that lowering the basicity below par level compromised 5-HT_6_R affinity very slightly, although it enhanced the selectivity. The X-ray structure analysis with 5-HT_6_R homology of **24i** and **25b** (Figure 14) revealed the binding mode, which further evidenced the in vitro experiment results. The results from these experiments concluded that 2-amidoimidazole-based moieties incorporated with other pharmacophoric features could provide new insights and lead to the development of new ligands of aminergic receptors as potential drug molecules [131].

Bromidge et al. reported novel tricyclic benzoxazine derivatives (**26a**–**z**), which were designed via bioisosteric replacement of the metabolically prone N-methyl amide group with comparatively smaller heterocyclic moieties to give these new tricyclic benzoxazines. These novel compounds were also found to be potent 5-HT1A/B/D receptor antagonists with some 5-HT transporter activity. In this study, compound **26d** (Figure 15) emerged as a prominent lead molecule, having 5-HT_1A/B/D_ receptor antagonists with zero intrinsic activity and K_i_ values of 9.5, 8.8, and 9.8, respectively, with 7.5 against hSerT. This lead compound, **26d**, displayed high selectivity over hERG potassium channels and a favorable pharmacokinetic profile during in vivo studies in the PD model. With a potentially balanced profile and encouraging results, compound **26d** was used by this research group as a clinical candidate for further investigations as a quick-acting antidepressant/anxiolytic with minimal to no chances of adverse/side effects [132].

## 5. Imidazole-Based Drugs under Clinical Trials against Depression

There are various imidazole-based drug candidates in different clinical trials for depression. These are listed in Table 1, with clinical trial numbers per data available at clinicaltrials.gov. EVT101 is an orally active NMDR antagonist that entered a phase II clinical trial but was terminated in the initial phase, and the FDA put this drug on hold for further development [133]. Pentoxifylline is a PDE inhibitor that has completed various phase I/II trials as an adjuvant in major depressive disorders [134]. In preclinical studies, it has shown promising results, and pretreatment of pentoxifylline itself showed antidepressant-like activity in animal models [135,136]. Etomidate is well explored for its effects on major depressive disorders in electroconvulsive therapies [137]. It completed a study on its role as a neurorestorative agent (NCT02667353), and one clinical phase IV study status is unknown (NCT02924090), in which it was being explored for major depression. Leuprolide acetate is known for its diverse pharmacological activities [138]. It has completed phase II (NCT04051320) and phase IV (NCT01116401) clinical trials against perinatal depression and menopause depression, respectively.

RO4917523, also known as basimglurant (2-chloro-4-[1-(4-fluoro-phenyl)-2,5-dimethyl-1H-imidazol-4-ylethynyl]-pyridine), is a negative allosteric modulator of the mGlu5 receptor under clinical trials for development against depression [139]. It has completed phase II trials (NCT00809562, NCT01437657) and is under further investigation by Hoffmann-La Roche for treatment-resistant and major depressive disorders. Candesartan is an angiotensin receptor blocker mainly used for hypertension treatment but has also been found to reverse depression-like symptoms in preclinical studies in the initial phase [140,141]. But, in clinical trials, it was withdrawn from phase I in the early phase (NCT04430959), and phase IV studies were terminated due to the adverse effects in treated patients (NCT01794455). Its further development as an antidepressant is still questionable. Cimicoxib, one of the NSAIDs used for pain relief and inflammation in dogs, has completed a phase II trial in major depressive disorder patients (NCT00510822) conducted by Affectis Pharmaceuticals AG [142]. The study was based on the hypothesis that adjuvant COX-2 inhibition can help reverse major depressive symptoms in patients. Valganciclovir, one of the antiviral compounds, is under phase I/II clinical trial for the antiviral treatment of cytomegalovirus in depression (NCT04724447). However, various severe side effects associated with this drug are major hurdles in further development [143]. Another imidazole derivative, Midazolam, a known benzodiazepine medicine, is under various trials for comparison in depressive patients (NCT05026203, NCT02360280, NCT05383313, NCT05528718, NCT01700829), although its role in treatment is still under question [144]. Thus, it is mostly being used as a control group for comparison of other medicines and their effects [145,146].

## 6. Recently Granted Patents on Imidazole-Based Compounds

Due to their unique structural characteristics and electron-rich environment, imidazole-containing molecules have attracted the attention of several research groups. It led to the development of various fused/unfused molecules with a wide range of bioactivities. Because of its wide variety of actions, many patents have been granted in the last few years. The patents for imidazole derivatives with antidepressant activity are listed in Table 2.

After reviewing the patents related to imidazole-based compounds, it is clear that in the last two decades, very few patents have been granted related to imidazole-based compounds for their potential against depression. Thus, comprehensive options are available to be explored in the upcoming time to explore this moiety for developing new antidepressant molecules with more significant potential and better efficacy.

## 7. Conclusions and Future Perspective

One key issue that has been unaddressed for a long time is finding a rational design with high preferential selectivity for serotonergic isoforms. Researchers attempted to gain preferential selectivity by bringing specific fused heterocyclic structures, monosaccharides, macrocycles, glycopeptides, and dendrimer-based scaffolds [152,153,154,155,156,157]. However, similar scaffolds with similar modifications exhibit multitarget activities, severely limiting their direct clinical implications [158,159,160,161,162,163]. Therefore, there is considerable demand for identifying newer chemical modalities with enough structural features to incorporate into pharmacophores to bring about selective serotonergic activities.

Due to its various pharmacological properties, the imidazole moiety has emerged as a significant pharmacophoric characteristic in medicinal chemistry. In this review, we aim to discuss the SAR analyses of several imidazole derivatives as antidepressants that target serotonin (5-HT) receptors. We compiled some studies and summarized the structure–activity relationship of the fundamental nucleus in all of them. We have developed a pharmacophore from existing selective serotonin reuptake inhibitors (SSRIs) on the open web server Pharmit. The pharmacophore obtained from the server contains five features: two aromatic rings in the violet color sphere, two hydrogen bond acceptors in an orange sphere, and hydrophobic in the green sphere (Figure 16).

Talking about imidazole acceptability in other chemical biology-oriented approaches, it has shown tremendous potential. These include serving as pH-responsive potent antimicrobials [164], improving the photophysical characteristics of azo dyes [165,166,167], decreasing side toxicities [13,168,169,170], and successful chemical integration in polymeric materials as theragnostic applications (such as in zeolitic imidazolate framework applications in drug delivery) [171,172,173,174,175,176,177,178,179,180].

After reviewing the data, it was evident that the imidazole moiety plays a critical role in treating depression by acting on serotonin receptors. The literature survey revealed that imidazole is a less explored moiety in developing 5-HT receptors that modulate chemical entities for treating depressive disorders. The molecules being explored against 5-HT-bearing imidazole are still in their initial states. This chemical feature, along with encouraging pharmacokinetic features, can provide broad scope for further exploration. Computer-based drug design can further speed up the imidazole-based drug development process with target specificity and efficiency. The electron-rich feature of this five-membered chemical moiety can help it bind with serotonin receptors and transporters very quickly. This can be beneficial for improving the efficacy of molecules [181]. The development of imidazole-based molecules against various disease states has become a hot topic [22,42,182,183,184]. This moiety can further enter into the development of antidepressants with its enriched structural features. This review may be used to draw key conclusions regarding the imidazole nucleus and appropriate substitutes for it, which will aid in developing novel antidepressant medications based on the imidazole nucleus.

## Figures and Tables

**Figure 1 pharmaceutics-15-02208-f001:**
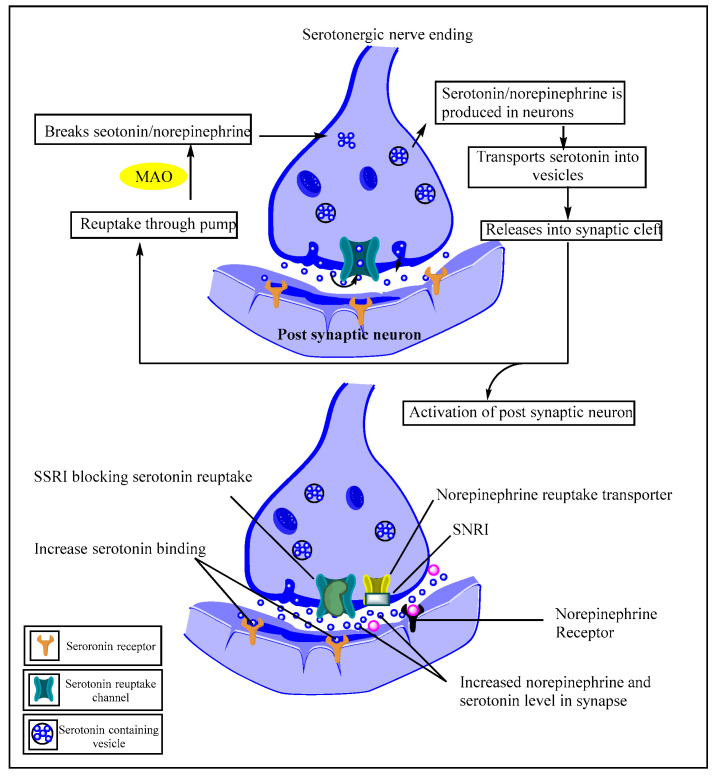
Role of serotonin receptor modulators in depression.

**Figure 2 pharmaceutics-15-02208-f002:**
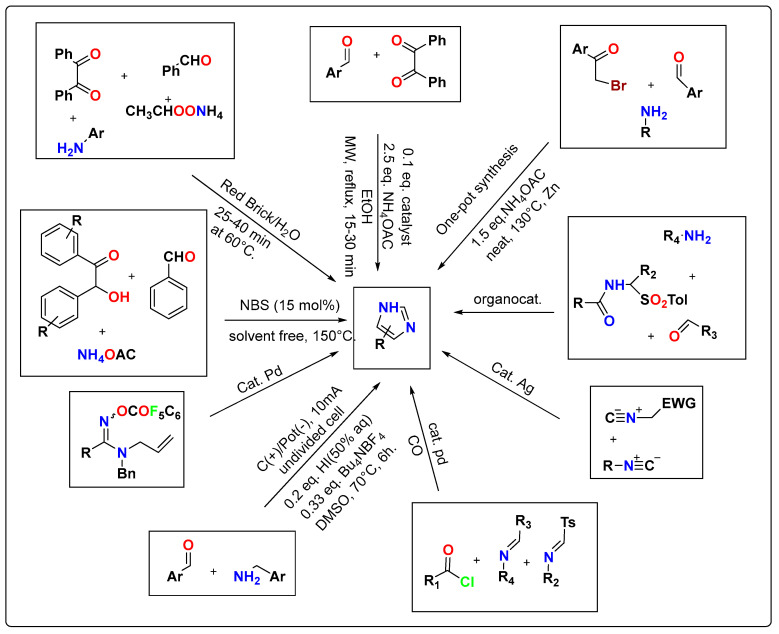
Recently developed synthetic methodologies for imidazole and its derivatives.

**Figure 3 pharmaceutics-15-02208-f003:**
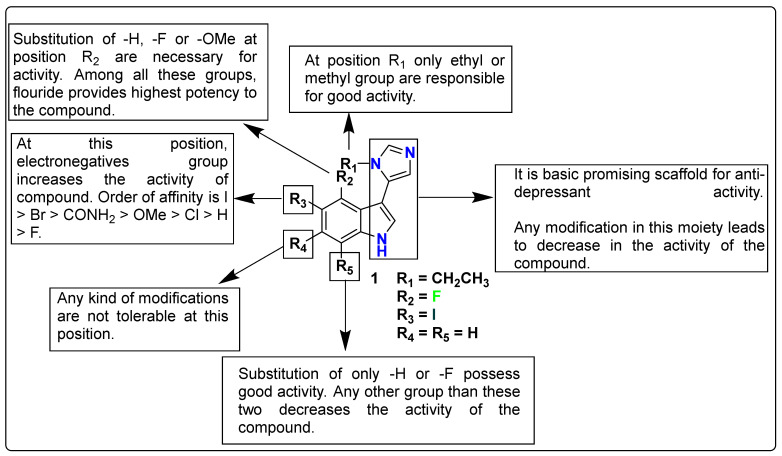
SAR studies of fluorinated 3-(1-alkyl-1H-imidazol-5-yl)-1H-indoles.

**Figure 4 pharmaceutics-15-02208-f004:**
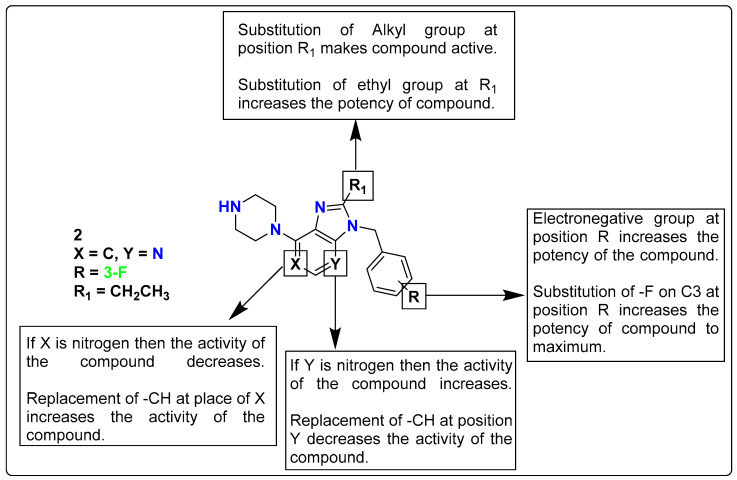
SAR studies of 3H-imidazo[4,5-*b*] pyridine and 3H-imidazo[4,5-*c*] pyridine derivatives.

**Figure 5 pharmaceutics-15-02208-f005:**
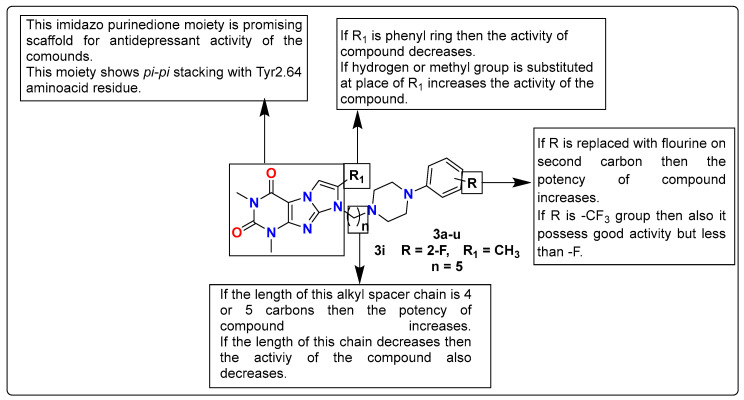
SAR studies of trifluoromethylphenylpiperazinylalkyl derivatives of 1H-imidazo[2,1-*f*]purine-2,4(3*H*,8*H*)-dione.

**Figure 6 pharmaceutics-15-02208-f006:**
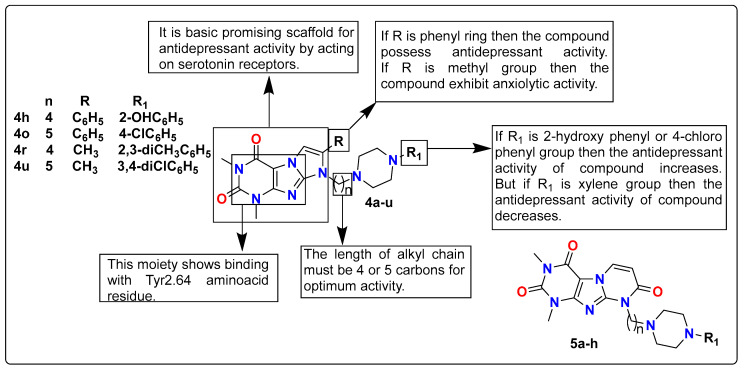
SAR studies of arylpiperazinylalkyl purine-2,4-diones.

**Figure 7 pharmaceutics-15-02208-f007:**
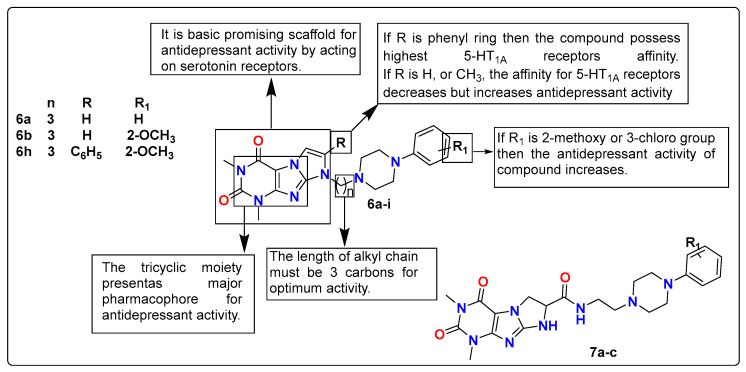
SAR studies of imidazo[2,1-*f*]purine-2,4-dione derivatives.

**Figure 8 pharmaceutics-15-02208-f008:**
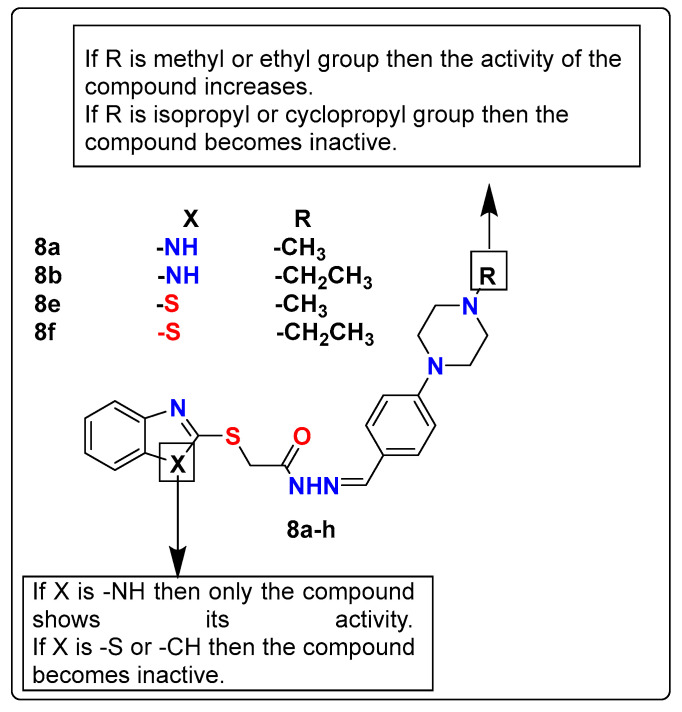
SAR studies of benzazole derivatives.

**Figure 9 pharmaceutics-15-02208-f009:**
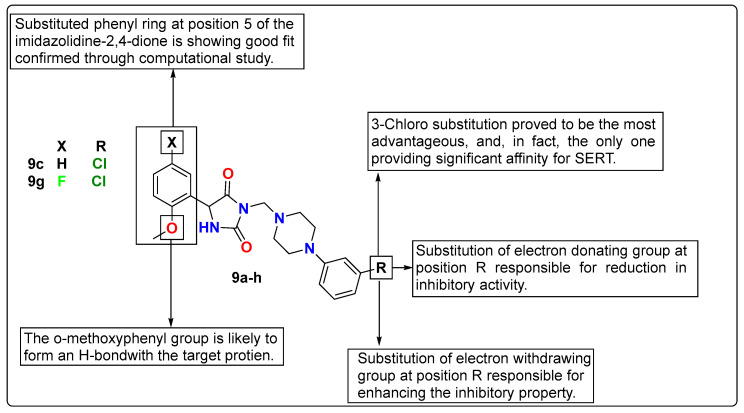
SAR studies of imidazolidine-2,4-dione derivatives.

**Figure 10 pharmaceutics-15-02208-f010:**
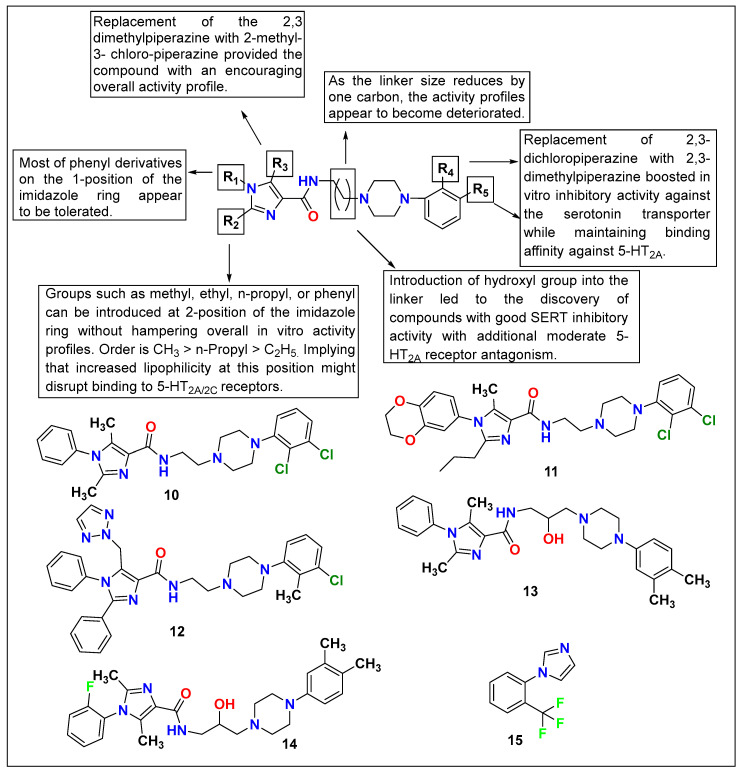
SAR studies of imidazole core containing arylpiperazine-4-carboxamide.

**Figure 11 pharmaceutics-15-02208-f011:**
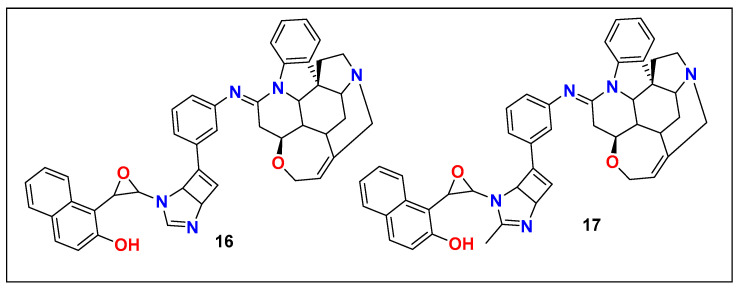
Strychnidin-oxiran-napthalenol derivatives.

**Figure 12 pharmaceutics-15-02208-f012:**
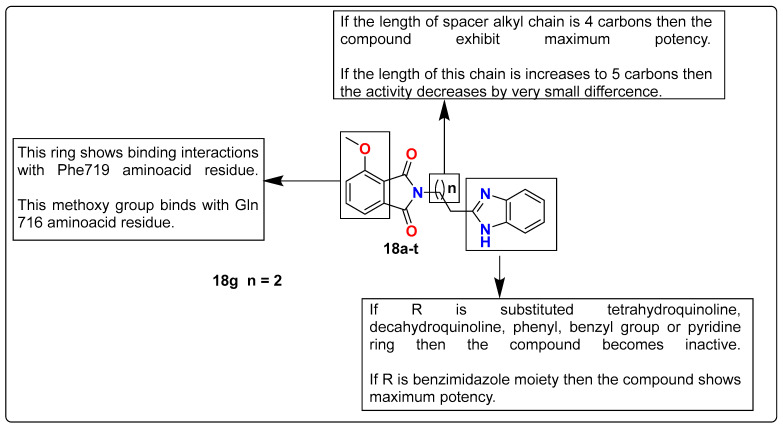
SAR studies of imidazole-containing 4-methoxy-1H-isoindole-1,3(2H)-dione derivatives.

**Figure 13 pharmaceutics-15-02208-f013:**
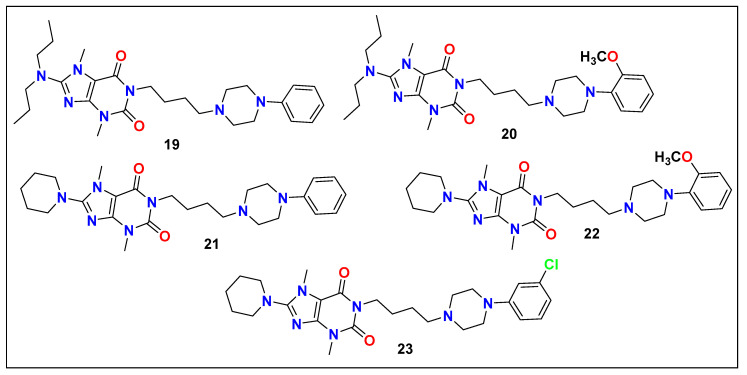
Structure of arylpiperazynylalkyl derivatives of 8-amino-3,7-dimethyl-1H-purine-2,6(3H,7H)-dione.

**Figure 14 pharmaceutics-15-02208-f014:**
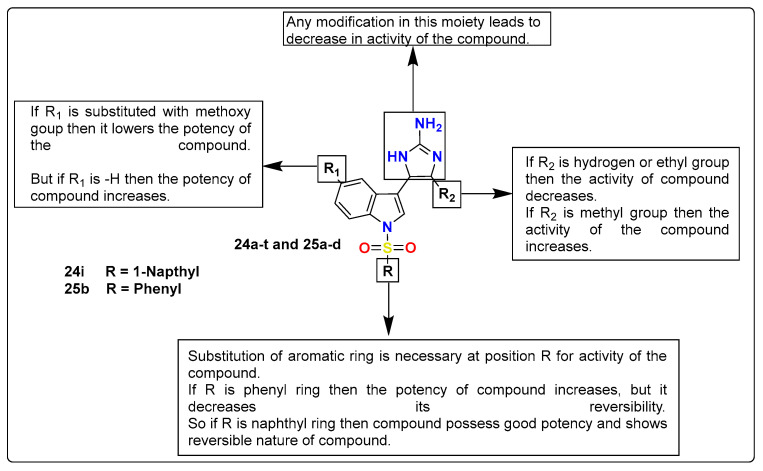
SAR studies of imidazole N-(1*H* imidazol-2-yl)acylamide derived imidazole derivatives.

**Figure 15 pharmaceutics-15-02208-f015:**
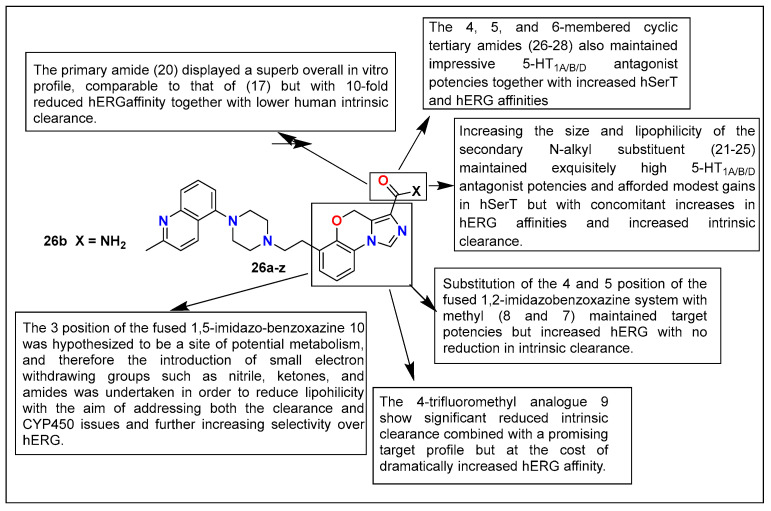
SAR studies of novel tricyclic benzoxazine derivatives.

**Figure 16 pharmaceutics-15-02208-f016:**
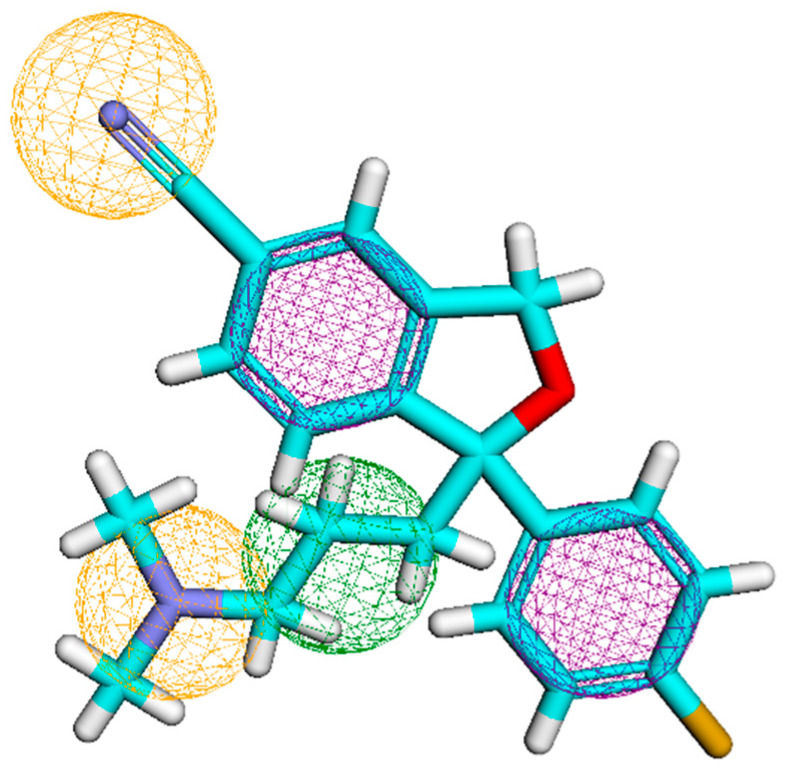
Pharmacophore generated using open web server Pharmit from existing SSRIs. It represents the presence of two aromatic rings (violet sphere), two hydrogen bond acceptors (orange sphere), and a hydrophobic region (green sphere).

**Table 1 pharmaceutics-15-02208-t001:** Imidazole-based drugs under clinical trials against depression.

S. No.	Compound	Structure	Clinical Trial Status	Clinical Trial Number
1	EVT101	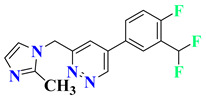	Phase 2 Terminated	NCT01128452
2	Pentoxifylline	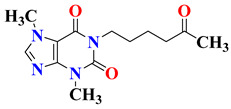	Phases 1 and 2 Completed	NCT05324735
			Early Phase 1 Completed	NCT04417049
			Not Applicable Completed	NCT03554447
			Phase 1 Completed	NCT05271084
3	Etomidate	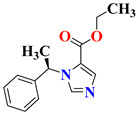	Not Applicable Completed	NCT02667353
			Phase 4 Unknown	NCT02924090
4	Leuprolide Acetate		Not Applicable Completed	NCT01762943
			Phase 2 Completed	NCT04051320
			Phase 4 Completed	NCT01116401
5	RO4917523	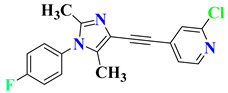	Phase 2 Completed	NCT00809562
			Phase 2 Completed	NCT01437657
6	Candesartan	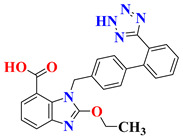	Early Phase 1 Withdrawn	NCT04430959
			Phase 4 Terminated	NCT01794455
7	Cimicoxib	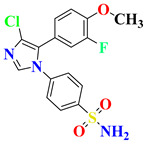	Phase 2 Completed	NCT00510822
8	Valganciclovir	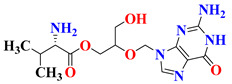	Phases 1 and 2 Recruited	NCT04724447
9	Midazolam	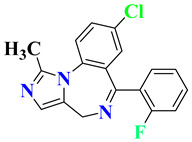	Phase 1 Terminated	NCT04082858
			Phase 4 Recruiting	NCT05026203
			Phase 2 Completed	NCT02360280
			Phase 2 Recruiting	NCT05383313
			Not yet recruiting	NCT05528718
			Phase 4 Completed	NCT01700829
			Phase 3 Active, not recruiting	NCT03889756
			Phase 4 Recruiting	NCT04220125
			Phase 3 Recruiting	NCT04939649

**Table 2 pharmaceutics-15-02208-t002:** Recently granted patents on imidazole-based compounds with antidepressant activities.

Patentee Name/Inventors	Title	Patent No.	Year	Ref.
Albert K, Bin L, Ray MD, Michael SJ, and Deyi Z	Imidazole carboxamides	Indian patent No. 271995	2016	[147]
Lee J, Seo HJ, Kang SY, Park EJ, Kim MJ, Lee SH, Kim JY, Kim J, Jung ME, Kim HJ, and Kim MS	Arylpiperazine-containing imidazole 4-carboxamide derivatives and a pharmaceutical composition comprising the same	US 8,835,436	2014	[148]
Ceccarelli SM, Jagasia R, Jakob-roetne R, and Wichmann J	Benzimidazoles as CNS active agents.	US 20,150,203,472A	2014	[149]
Schwartz JC and Lecomte JM	Combination product comprising an antagonist or inverse agonist of histamine receptor H3 and an antipsychotic and antidepressant agent, and use thereof for the preparation of a medicament that prevents the adverse effects of psychotropic drugs.	US 8,106,041	2012	[150]
Thurkauf A, Horvath RF, Yuan J, and Peterson JM	Certain 4-aminomethyl-2-substituted imidazole derivatives and 2-aminomethyl-4-substituted imidazole derivatives; new classes of dopamine receptor subtype-specific ligands.	US 6,797,824	2004	[151]

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
