# Peer review of "Imidazoles as Serotonin Receptor Modulators for Treatment of Depression: Structural Insights and Structure–Activity Relationship Studies"

_pharmaceutics, 2023, doi:10.3390/pharmaceutics15092208_

Round 1

Reviewer 1 Report

The manuscript is a result of important work and evaluation of the actual knowledge in the field. In the same time, even useful, it looks more like a book chapter than an article. For the reader it could be better organize in a simplify style, with more data in „supplementary material” readable by the interested readers.
It would be useful to apply to an evaluation regarding English topic, wording and phrases organization.
It should read again an correct any typos and small mistakes (e.g. et all with italics and without italics, bold where it should be not etc.)
Please clarify the source of the figures. Are they original? Are they adapted?

The same comments as above. Maybe, a professional evaluation for English style and quality is needed.

Author Response

Reviewer 1 comments:

The manuscript is a result of important work and evaluation of the actual knowledge in the field. In the same time, even useful, it looks more like a book chapter than an article. For the reader it could be better organize in a simplify style, with more data in supplementary material” readable by the interested readers.

Reply: The review article is modified for the better understanding and enhancing information in the manuscript.

It would be useful to apply to an evaluation regarding English topic, wording and phrases organization.

Reply: The manuscript is revised for grammer and phrasing again thoroughly.

It should read again an correct any typos and small mistakes (e.g. et all with italics and without italics, bold where it should be not etc.)

Reply: The manuscript is revised for graphical and typographical errors and changes are mentioned as track changes.

Please clarify the source of the figures. Are they original? Are they adapted?

Reply: The figures are all original and created by authors ourself.

Reviewer 2 Report

In the present work, the authors report the review regarding imidazole-bearing pharmacophores as the inhibitors of serotoninergic signaling. The manuscript is interesting and deserves to be published after minor correction as follows:
- Introduction section lacks a link between the second and third paragraphs. I suggest adding general information on serotonin modulators other than imidiazoles and discarding information on the unrelated wiyh manuscript title functions of imidiazoles. In addition, it would make it easier for readers to provide a general formula for this class of compounds. I am not an expert in chemical synthesis but I would like to highlight the rest of the manuscript, which reads well. In addition, the inclusion of two tables summarizing the clinical use of imidiazoles is a very good idea.

Author Response

Reviewer 2 comments

In the present work, the authors report the review regarding imidazole-bearing pharmacophores as the inhibitors of serotoninergic signaling. The manuscript is interesting and deserves to be published after minor correction as follows:

- Introduction section lacks a link between the second and third paragraphs. I suggest adding general information on serotonin modulators other than imidiazoles and discarding information on the unrelated with manuscript title functions of imidiazoles.

Reply: The manuscript is revised for improving the paragraphs and information.

In addition, it would make it easier for readers to provide a general formula for this class of compounds.

Reply: The suggested changes are incorporated and general pharmacophore is added in conclusion section.

I am not an expert in chemical synthesis but I would like to highlight the rest of the manuscript, which reads well. In addition, the inclusion of two tables summarizing the clinical use of imidiazoles is a very good idea.

Reply: The authors are thankful to the reviewer for evaluation and appreciation of our work. We extend our thanks for valuable time and comments also.

Reviewer 3 Report

The authors Goel et al. have reported the synthetic methodology, recent advancements and SAR studies of Imidazole derivatives as antidepressants targeting Serotonin receptor modulators. The manuscript is well organized, and the results are well explained for the easy understanding of the audience. The authors have also compiled a list of various imidazole derivatives under clinical trials for depression. The work is of utter importance to the community and acceptable for publication in this reputed journal. The manuscript can be accepted with the following minor corrections.

·       Authors have disclosed pictorially the role of serotonin receptor modulators in depression. However, insights into the serotonin receptor's binding site and key amino acids in the active domain are missing. This will enhance the medicinal chemistry-based aspect of the present work.

·       In the introduction, section authors should also discuss brief history and case study of serotonin discovery and how it has brought about a revolution in the branches of neuropharmacology and associated drug discovery.

·       The author should also conclude on an ideal pharmacophore that could plausibly act as a Serotonin receptor antagonist based on the insights drawn from the present review of the literature. 

·       Chemical structure size needs to be uniform throughout the manuscript.

·       Fig. 7: “If R is H or CH3……” need to be revised.

·       The manuscript is needed to be revised for minor grammatical errors.

Author Response

Reviewer 3 comments:

Authors have disclosed pictorially the role of serotonin receptor modulators in depression. However, insights into the serotonin receptor's binding site and key amino acids in the active domain are missing. This will enhance the medicinal chemistry-based aspect of the present work.

Reply: The manuscript is revised and new section is added after introduction for structural insights of serotonin receptor.

In the introduction, section authors should also discuss brief history and case study of serotonin discovery and how it has brought about a revolution in the branches of neuropharmacology and associated drug discovery.

Reply: The history of discovery and study of serotonin is added in introduction in the revised manuscript.

The author should also conclude on an ideal pharmacophore that could plausibly act as a Serotonin receptor antagonist based on the insights drawn from the present review of the literature. 

Reply: The suggested changes are incorporated in the revised manuscript and pharmacophore is added in conclusion section.

Chemical structure size needs to be uniform throughout the manuscript.

Reply: The structures are revised for uniformity.

Fig. 7: “If R is H or CH3……” need to be revised.

Reply: The changes are made and marked in yellow.

The manuscript is needed to be revised for minor grammatical errors.

Reply: The manuscript is revised for grammer and changes are mentioned in track changes in revised file.
